# Primary Bone Marrow HIV-Associated Hodgkin Lymphoma Complicated by Hemophagocytic Lymphohistiocytosis

**DOI:** 10.3390/genes13091608

**Published:** 2022-09-08

**Authors:** Ji Lin, Binsah George

**Affiliations:** 1Department of Internal Medicine, The University of Texas Health Science Center at Houston, Houston, TX 77030, USA; 2Department of Hematology/Oncology, The University of Texas Health Science Center at Houston, Houston, TX 77030, USA

**Keywords:** primary bone marrow Hodgkin lymphoma, hemophagocytic lymphohistiocytosis, HIV-associated Hodgkin lymphoma

## Abstract

The presentation of human immunodeficiency virus (HIV)-associated Hodgkin lymphoma can differ from that of the general population. More specifically, primary bone marrow Hodgkin lymphoma is an uncommon presentation that is more often reported in patients with HIV. Given the many overlapping symptoms of Hodgkin lymphoma and HIV as well as HIV-associated infections, diagnosis can be difficult and delayed. We describe a case of primary bone marrow HIV-associated Hodgkin lymphoma complicated by hemophagocytic lymphohistiocytosis (HLH) where the initial work-up was inconclusive. Our case demonstrates the importance of early consideration of HLH as well as the need for an early bone marrow biopsy in a cytopenic patient with a fever of unknown origin.

## 1. Introduction

Human immunodeficiency virus (HIV)-associated Hodgkin lymphoma (HL) differs from HL in the general population in that HIV-associated HL more frequently presents with B symptoms and has more extensive disease involvement [1,2]. Primary bone marrow-limited Hodgkin lymphoma is uncommon and mostly reported in HIV-positive patients [3,4]. Given the uncommon presentation and overlapping symptoms with HIV and HIV-associated infections, primary bone marrow HIV-associated HL is a difficult diagnosis. We describe a case of HIV-associated classic Hodgkin lymphoma with primary bone marrow involvement complicated by hemophagocytic lymphohistiocytosis (HLH) where the treatment was delayed due to difficulty in making the diagnosis. Although a bone marrow biopsy is not usually necessary for the diagnosis of HL, we suggest that it may be useful in patients with HIV presenting with cytopenia and fever despite the fact that these presentations are often attributed to HIV-related complications. 

## 2. Case Presentation

A 64-year-old male patient with a past medical history of HIV diagnosed seven years prior, prior pneumocystis jiroveci pneumonia treated with chronic trimethoprim/sulfamethoxazole (TMP-SMX) therapy, and no previous history of blood dyscrasias or malignancies except for a left lung nodule that was biopsied and found negative for malignancy presented to the emergency department from the HIV clinic for six months of fatigue, dyspnea on exertion, as well as for one week of fever and night sweats. The patient has chronic leukopenia associated with HIV, but was found to have worsening anemia with a hemoglobin of 4.6 g/dL from a baseline of 10 g/dL and new thrombocytopenia with a platelet measuring 82 × 103 µL. The patient was admitted for further work-up. The most recent CD4 count was 8 cells/ µL from the previous baseline of 66–97 cells/µL, despite reporting compliance with bictegravir/emtricitabine/tenofovir antiretroviral therapy (ART) and having an undetectable HIV viral load. Most likely, differentials at that time included medication-induced pancytopenia from TMP-SMX, HIV-induced cytopenia, infection, and HLH. In terms of a differential diagnosis of HLH, he met 5 out of 8 criteria for fever, hyperferritinemia of 4688 ng/mL, splenomegaly of 21.1 cm, peripheral blood cytopenia, and hypertriglyceridemia of 332 mg/dL, suggesting a 96–98% probability of HLH; however, an infection was still thought to be more likely as the underlying etiology of the HLH, especially since the CT scan did not show any evidence of extensive lymphadenopathy but continued to demonstrate chronic splenomegaly with portal hypertension with concurrent findings suggestive of cirrhosis, possibly due to nonalcoholic fatty steatohepatitis or a previously resolved hepatitis B infection. For the above reasons, the splenomegaly was not considered secondary to the HLH, but rather to be caused by the underlying portal hypertension and cirrhosis. The infectious work-up showed that parvovirus B19, cytomegalovirus, hepatitis C virus, and herpes virus type 8 were negative, but Epstein–Barr virus (EBV) DNA was positive at 51 U/mL. Acid-fast bacilli and Grocott methenamine silver stains of the bone marrow biopsy for bacterial and fungal culture, including the histoplasmosis antigen, were also negative. Six months prior to this admission, the patient had a PET scan for the work-up of a suspicious lung nodule which also showed a left external iliac lymph node less than one centimeter in size. Since the initial PET scan, the lung nodule had been biopsied twice with an 18-gauge core biopsy, with four samples taken each time and specimen adequacy confirmed by pathology; it was negative for malignancy each time. Given the new concern for lymphoma, a lymph node biopsy of the left external iliac lymph node was pursued, but the lymph nodes were deemed too small for biopsy by the interventional radiology and general surgery. A PET scan during this admission was not pursued since the CT scan on the current admission was stable compared to previous PET-CT with no new evidence of lymphadenopathy or any suspicious lesions. Hence, an underlying lymphoma or myeloproliferative pathology was deemed unlikely. A bone marrow biopsy, which was carried out to assist with the diagnosis of HLH, showed no evidence of hemophagocytic lymphohistiocytosis, but was positive for EBV, CD30, and large atypical lymphoid cells suggestive of EBV+ classic Hodgkin lymphoma (HL). However, the flow cytometry of the bone marrow sample showed no clonality for lymphoproliferative or leukemic lineage. After further evaluation, HLH was deemed unlikely given the chronicity of the patient’s symptoms, clinical and laboratory improvement, and resolution of fever with supportive therapy alone. Many of the criteria for HLH was attributed to the patient’s active HIV status and EBV infection. Thus, the patient was discharged with a close hematology follow-up. 

When the patient followed up in the hematology clinic approximately 2 weeks later, he was noted to still have persistent pancytopenia, but with a new fever; therefore, he was readmitted for a further work-up. The most likely differential was thought to be HLH or splenic lymphoma. The initial work-up included obtaining soluble IL-2 receptors for the consideration of a splenic biopsy and rebiopsy of the bone marrow. Meanwhile, infectious disease was reconsulted with recommendations to work-up the disseminated Mycobacterium avium complex given the elevated alkaline phosphate after a therapeutic trial of antibiotics. The rheumatologic work-up was also sent, which included antinuclear antibodies, the rheumatoid factor, anti-cyclic citrullinated peptides, the erythrocyte sedimentation rate, and C-reactive proteins, which eventually came back negative. No other treatment was started at this time given that his clinic picture was relatively mild and chronic without a need to initiate emergent therapy. The Il-2 receptor level returned at 27 U/mL (Figure 1) and the EBV level was now increased to 148 U/mL. The splenic biopsy was deemed unsafe by interventional radiology. The bone marrow biopsy was repeated, again, looking for evidence of hemophagocytic lymphohistiocytosis. The repeat bone marrow biopsy was largely replaced by tumor cells with typical, large, and pleomorphic cells with a prominent cherry red nucleoli, high nuclear-cytoplasmic ratio, and irregular nuclear contours. Reed–Sternberg cells were present with both binucleated and multinucleated forms seen. Immunohistochemical stains showed CD15 to be patchy positive, CD30 to be diffusely positive, and EBV to be positive. Altogether, this confirmed a diagnosis of EBV-associated classic Hodgkin lymphoma, Lugano stage IV, complicated by HLH. The patient was initially treated for HLH with two doses of etoposide and dexamethasone per the HLH-94 protocol. Intrathecal chemotherapy was not warranted as the patient did not show any signs or symptoms of central nervous system involvement. The patient received rituximab as an EBV-directed therapy, along with HL-directed therapy through two lead-in doses of single-agent brentuximab vedotin (BV) at 1.2 mg/kg (reduced dose due to abnormal liver function tests) IV once every three weeks followed by doxorubicin, vinblastine, and dacarbazine (AVD) for six cycles and consolidative BV for four cycles [5]. His Hodgkin Lymphoma International Prognostic Score at diagnosis was 6. After tolerating two doses of etoposide and a single dose of BV, the patient was discharged. PET-CT after two cycles of BV showed no FDG-avid lymphadenopathy. At the subsequent follow-up, the patient reported feeling the best he has felt in a long time.

## 3. Discussion

Hodgkin lymphoma classically presents as lymphadenopathy around the head, neck, axilla, or mediastinum. Constitutional symptoms such as fever and fatigue may also be present weeks to months before evaluation. Diagnosis is typically made by microscopic findings with immunophenotypes for Hodgkin/Reed–Sternberg cells. The presentation of HL differs slightly in HIV patients compared to the general population, making the diagnosis difficult which can lead to a delay in care. We described a case of HIV-associated classic Hodgkin lymphoma with primary bone marrow involvement complicated by HLH where the treatment was delayed due to difficulty in making the diagnosis. 

HIV-associated HL differs from HL in the general population due to the widespread extent of the disease at presentation. In the general population, about 38% of patients present with B symptoms at presentation compared to 70–96% in HIV-associated HL [1,2]. Furthermore, 74–92% of patients have extranodal involvements, with common sites being bone marrow, liver, and spleen [2]. Primary bone marrow-limited Hodgkin lymphoma is uncommon and mostly reported in HIV-positive patients, although some cases have been reported in HIV-negative patients as well [3,4]. The presentation of primary bone marrow-limited Hodgkin lymphoma also differs from HIV-associated HL to the point that some question whether it may be its own entity [6]. In a retrospective study by Ponzoni et al., it was found that 14% of patients with newly diagnosed HIV-associated Hodgkin lymphoma had bone marrow as the only site of disease. Given the uncommon presentation and many overlapping symptoms between lymphoma and HIV and HIV-associated infections, this can lead to a delay in diagnosis and care since infections are generally more common. 

The incidence of HL has increased since the introduction of ART [7]. This has been speculated to be due to the fact that a certain number of CD4+ lymphocytes are needed to create the microenvironment for the development of Hodgkin/Reed–Sternberg cells. In addition to the difference in initial presentation between HIV-associated HL and HL in the general population, HIV-associated HL is more often associated with EBV infection [2,7]. Under morphological examination, in HIV-associated HL there is a greater predominance of Hodgkin/Reed–Sternberg cells compared to in HL in the general population [2,7]. Furthermore, HIV-associated HL is more often characterized histologically by the mixed cellularity and lymphocyte-depleted cell type, which are deemed unfavorable [2]. Despite this, HIV-associated HL has a similar response rate and survival compared to HL in the general population when treated with the same therapy [7].

During the diagnosis of HL, a bone marrow biopsy is generally not needed due to the reliability of PET scans in diagnosing bone marrow infiltration [7]. In our case, the PET scan six months prior showed a lung nodule that had been biopsied twice and found to be negative for malignancy each time and a stable left external iliac lymph node that was considered too small to biopsy. A bone marrow biopsy was performed to assist with the diagnosis of HLH and any possible underlying bone marrow etiology. The findings were suspicious but not compelling for HL, and the flow cytometry was inconclusive [8]. It was not until the second bone marrow biopsy, again carried out to aid in the underlying etiology of HLH, that a diagnosis of classic Hodgkin lymphoma was made. 

Our patient’s HIV-associated HL was complicated by secondary HLH, for which the diagnosis was also difficult to make. HLH is a rare syndrome caused by the uncontrolled activation of the immune system. Although the patient met five out of eight of the HLH-2004 criteria, a diagnosis of HLH in a patient with HIV is difficult because of overlapping symptoms. Some reports suggest that HLH-2004 may not be of benefit in patients with sepsis such as our patient [9]. Raschke et al. suggest that progressive pancytopenia may be a feature that is more likely to suggest secondary HLH as opposed to septic shock, but in our case, this has shown itself to not be useful in patients with chronic cytopenia such as a patient with HIV. Further complicating the picture is the fact that ferritin, which is an inflammatory marker and elevated in both infection and HLH, has also been shown to be markedly elevated in patients with untreated classic Hodgkin lymphoma without lymphadenopathy [10].

Otrock et al. found that the most common triggers of HLH were attributed to infection, malignancies, autoimmune disorders, primary immunodeficiency, and post-solid organ transplant [11]. Although our patient had HIV and EBV, two viruses known to trigger HLH, the diagnosis was not made initially due to the fact that he got clinically better with supportive care alone and without the treatment of etoposide and dexamethasone. Interestingly, the EBV viral load has been correlated with HLH severity and during our patient’s second admission, when the diagnosis of HLH was apparent, his EBV viral load increased from 51K to 148K [12]. This is likely because EBV plays an etiological role in the pathogenesis of HIV-associated HL [2]. Furthermore, as seen in Figure 1, our patient’s IL-2 receptor decreased after starting on treatment. The IL-2 receptor tends to be a reliable lab for the monitoring of HLH activity and response to therapy, whereas ferritin may not be as reliable [13]. Of note, although our patient did not show hemophagocytic lymphohistiocytosis on both bone marrow biopsies, it is not needed to make the diagnosis per the HLH-2004 criteria. About 70% of those with HLH syndrome present with features of hemophagocytic lymphohistiocytosis through bone marrow aspiration, and some studies suggest that it may aid in the diagnosis if there is uncertainty [8,14]. In our patient, HLH was likely triggered by a combination of EBV, HIV, and HL. 

## 4. Conclusions

The diagnosis of HL is difficult in patients with HIV due to the often atypical presentation. Diagnosis of HLH is also difficult due to overlapping symptoms with HIV and HIV-associated infections. Our case demonstrates the importance of the early consideration of HLH and possible underlying HL in a cytopenic patient with a fever of unknown origin in the setting of HIV. The early detection of HLH can lead to a quick treatment, but also a thorough work-up for secondary causes [15]. Furthermore, although the role of a bone marrow biopsy has been challenged in the work-up of HL in favor of imaging, such as PET-CT, our case demonstrates that it may still have a role in the work-up of HIV patients for primary bone marrow HL [3,4].

## Figures and Tables

**Figure 1 genes-13-01608-f001:**
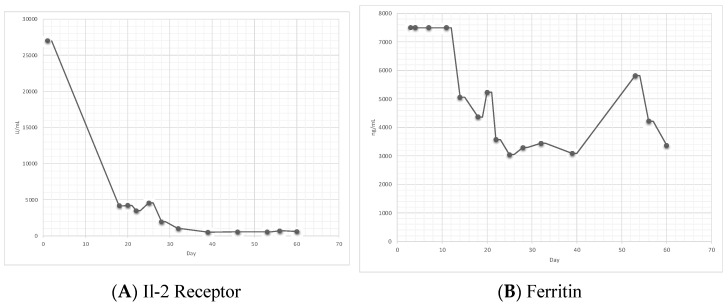
(**A**) shows the trend of IL-2 receptor alpha level. (**B**) shows trend of ferritin level. First dose of etoposide was given on day 7 and the second dose of etoposide was given on day 16. Of note, in (**B**), prior to day 13, ferritin level was greater than 7500 ng/mL and incalculable.

## Data Availability

Not applicable.

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
