# Peer review of "Primary Bone Marrow HIV-Associated Hodgkin Lymphoma Complicated by Hemophagocytic Lymphohistiocytosis"

_genes, 2022, doi:10.3390/genes13091608_

Round 1
Reviewer 1 Report
Thank you for the opportunity to review this interesting and well written manuscript about primary bone marrow HIV-associated lymphoma complicated by HLH. This study highlights the importance consider this differential diagnosis. Please consider the following comments:
Case Presentation
I believe it is important to provide more information about the patient's medical history.
I recommend providing information on the current antiretroviral treatment regimen, the baseline cd4 count. Knowing this information seems relevant since the study by Rajme S. et al (doi.org/10.1016/j.htct.2021.02.008) suggests that the time from HIV diagnosis to diagnosis of peripheral blood cytopenias greater than one year makes it more likely to be a malignant etiology.
I also consider it important to add to the medical history whether the patient had a history of blood dyscrasias or malignancies
Discussion
The authors might cite the relevant paper entitled “HIV-Associated Primary Bone Marrow Hodgkin’s Lymphoma: A Distinct Entity?” by Shah et al (10.1200/JCO.2010.28.3077) which is appropriate to this study.
Author Response
Thank you for the opportunity to review this interesting and well written manuscript about primary bone marrow HIV-associated lymphoma complicated by HLH. This study highlights the importance consider this differential diagnosis. Please consider the following comments:
Case Presentation
I believe it is important to provide more information about the patient's medical history.
I recommend providing information on the current antiretroviral treatment regimen, the baseline cd4 count. Knowing this information seems relevant since the study by Rajme S. et al (doi.org/10.1016/j.htct.2021.02.008) suggests that the time from HIV diagnosis to diagnosis of peripheral blood cytopenias greater than one year makes it more likely to be a malignant etiology.
I also consider it important to add to the medical history whether the patient had a history of blood dyscrasias or malignancies
The patient’s HIV was diagnosed 7 years prior. We have his baseline the prior two years he was in our system (66-97). This information has been included. Thank you for the suggestions.
Discussion
The authors might cite the relevant paper entitled “HIV-Associated Primary Bone Marrow Hodgkin’s Lymphoma: A Distinct Entity?” by Shah et al (10.1200/JCO.2010.28.3077) which is appropriate to this study.
We actually came across this article while reading more about this topic as well and found it very interesting. We’ve decided to include it in the discussion. Thank you for the suggestions

Reviewer 2 Report
Please use the term „hemophagocytic lymphohistiocytosis” instead of ” Hemophagocytosis lymphohistiocytosis”
Introduction
“Although a bone marrow biopsy is not usually necessary for the diagnosis of HL, we suggest that it may be useful in patients with HIV presenting with cytopenia and fever. “
Bone marrow biopsy is typically used for staging and not for diagnosis in HL. In the reported case it was used for diagnosis which cannot be compared with a typical situation. In the case of the diagnosis, the histopathological examination of the affected tissue is mandatory.
Please rephrase the sentence to make it clear to the reader.
There is nothing unusual in performing a bone marrow biopsy in a patient with cytopenia of an unknown reason.
Case presentation
Please use the appropriate units for all blood parameters reported in the text
e.g. Hgb is typically given in g/dL, PLT as x 109/L
please delete the word “leukopenia” – it is enough to report lymphopenia (please provide the value) if the patient did not suffer from neutropenia
please provide units for CD4 count
If the patient did not have neutropenia he should not be reported as having pancytopenia – please verify
Differential diagnosis: it is strange that DD did not include any hematological diseases
Please report how big the spleen was
Please provide the units for EBV
“Acid-fast bacilli and Grocott methenamine silver stains and fungal culture including Histoplasmosis antigen were also negative” – which material was tested?
“…the patient had a PET scan for work-up of a suspicious lung nodule which also showed a left external iliac lymph node less than one centimeter in size. Since the initial PET scan, the lung nodule has been biopsy twice and had been negative for malignancy each time.” – what was the Deauville score? What type of biopsy was performed? Fine needle? Core? – negative results do not exclude the possibility of HL; What about fungi – were they also tested?
“Given new concern for Hodgkin lymphoma, a lymph node biopsy..” – I do not understand why the Authors started suspecting HL; it would be more reasonable (if justified at all) to suspect a lymphoma than HL
“However, flow cytometry of bone marrow sample showed no clonality for lymphoproliferative or leukemic lineage.” – FC is not used for the diagnosis of HL!!! The inflammatory infiltration in HL is not clonal!!!
Please explain the abbreviations MAC, ANA, RF, CCP, ESR, CRP
“Immunohistochemical stains showed CD15 patchy positive, CD30 diffuse positive, and EBV positive making a diagnosis of Stage 4 EBV-associated classic Hodgkin lymphoma complicated by HLH.” – there is no such histopathological diagnosis; the diagnosis could be only classic HL – please provide the full description along with EBV tests; what were the signs of hemophagocytosis? Please rewrite the sentence
Stage 4 according to which classification?
Why did the patient start treatment with HLH protocol and not with HL-oriented chemotherapy? In the case of lymphoma-associated HLH the treatment should be lymphoma oriented
“PET-CT after two cycles…” – there is no information that PET/CT was positive before
Discussion
“Hodgkin lymphoma classically presents as lymphadenopathy around the head, neck, axilla, and mediastinum.” – it would be better to use “or” instead of “and”
“We described a case of HIV-associated Classic Hodgkin Lymphoma with primary bone marrow involvement complicated by HLH where the treatment was delayed due to difficulty in making the diagnosis.” – or because not making appropriate tests at an appropriate time?!
“HIV-associated HL are also characterized by the predominance of Hodgkin/Reed-Sternberg cells whereas they are less commonly found in HL in the general population.” – what do the Authors mean?
“During the diagnosis of HL, bone marrow biopsy is generally not needed due to the reliability of PET scans in diagnosing bone marrow infiltration.6 In our case, PET scan six months prior showed a lung nodule that had been biopsied twice and negative for malig-nancy each time and a stable left external iliac lymph node that was considered too small to biopsy. A bone marrow biopsy was done to assist with the diagnosis of HLH and any possible underlying bone marrow etiology. Findings were suspicious but not compelling for HL, and the flow cytometry was inconclusive.7” – please see the comments above – the whole paragraph is incorrect from the hematological point of view
“Some reports suggest that HLH-2004 may not be of benefit in patients with sepsis such as our patient” – I do not understand what the Authors mean – the patient was not diagnosed with sepsis
“Further complicating the picture is the fact that ferritin, which is an inflammatory marker and elevated in both infection and HLH has also been shown to be markedly ele-vated in patients with untreated classic Hodgkin lymphoma without lymphadenopathy.9” – ferritin alone is not used for diagnosing HLH – I cannot see any reason for such a sentence
“Interestingly, EBV viral load has been correlated with HLH severity and during our pa-tient’s second admission, when the diagnosis of HLH was apparent, his EBV viral load increased from 51K to 148K.11 This is likely because EBV plays an etiological role in the pathogenesis of HIV-associated HL.14” – the Authors mix HLH with HL – this does not make a logical sequence
Conclusions
Conclusions need to be modified after modification of the text
Author Response
Introduction
“Although a bone marrow biopsy is not usually necessary for the diagnosis of HL, we suggest that it may be useful in patients with HIV presenting with cytopenia and fever. “
Bone marrow biopsy is typically used for staging and not for diagnosis in HL. In the reported case it was used for diagnosis which cannot be compared with a typical situation. In the case of the diagnosis, the histopathological examination of the affected tissue is mandatory.
Please rephrase the sentence to make it clear to the reader.
There is nothing unusual in performing a bone marrow biopsy in a patient with cytopenia of an unknown reason.
We agree that bone marrow is not use for diagnosis. We meant to suggest that cytopenia in HIV patient is often dismissed as HIV progression/complications so hematologic dyscrasias are often missed. Hope this clears it up.
Case presentation
Please use the appropriate units for all blood parameters reported in the text
e.g. Hgb is typically given in g/dL, PLT as x 109/L
please delete the word “leukopenia” – it is enough to report lymphopenia (please provide the value) if the patient did not suffer from neutropenia
please provide units for CD4 count
Information included. Thank you for the suggestions.
If the patient did not have neutropenia he should not be reported as having pancytopenia – please verify
Differential diagnosis: it is strange that DD did not include any hematological diseases
We agree that hematological disease should be part of the differential. We meant to say that given history of HIV, HIV associated complications were highest on the differential at the time. Along with the fact that hematologic testing along with imaging done return more or less as expected, it made the diagnosis of primary bone marrow limited HL difficult.
Please report how big the spleen was
Information included. Thank you for the suggestions.
Please provide the units for EBV
Information included. Thank you for the suggestions.
“Acid-fast bacilli and Grocott methenamine silver stains and fungal culture including Histoplasmosis antigen were also negative” – which material was tested?
Information included. Thank you for the suggestions.
“…the patient had a PET scan for work-up of a suspicious lung nodule which also showed a left external iliac lymph node less than one centimeter in size. Since the initial PET scan, the lung nodule has been biopsy twice and had been negative for malignancy each time.” – what was the Deauville score? What type of biopsy was performed? Fine needle? Core? – negative results do not exclude the possibility of HL; What about fungi – were they also tested?
18-gauge core biopsy x4 for each biopsy, First biopsy was not stained for fungi but the second biopsy was stained for AFB and culture of which was negative. Lung SUV score of 5.7 and 2.6. Illiac lymph node SUV score of 4.6 and 8.8. Further information about biopsy included. Hope this makes it more clear.
“Given new concern for Hodgkin lymphoma, a lymph node biopsy..” – I do not understand why the Authors started suspecting HL; it would be more reasonable (if justified at all) to suspect a lymphoma than HL
Changed HL to lymphoma.Suspicion for lymphoma given reactive iliac lymph node on prior PET.
“However, flow cytometry of bone marrow sample showed no clonality for lymphoproliferative or leukemic lineage.” – FC is not used for the diagnosis of HL!!! The inflammatory infiltration in HL is not clonal!!!
We agree with your statement. However, in our case where the diagnosis of HL is less clear given previously non-suggestive PET and stable CT scan and now a bone marrow biopsy suggestive of HL. We think a FC is appropriate as an ancillary test.
Grewal, R. K., Chetty, M., Abayomi, E. A., Tomuleasa, C., & Fromm, J. R. (2019). Use of flow cytometry in the phenotypic diagnosis of hodgkin's lymphoma. Cytometry. Part B, Clinical cytometry, 96(2), 116–127. https://doi.org/10.1002/cyto.b.21724
Please explain the abbreviations MAC, ANA, RF, CCP, ESR, CRP
HL – please
Information included. Thank you for the suggestions.
“Immunohistochemical stains showed CD15 patchy positive, CD30 diffuse positive, and EBV positive making a diagnosis of Stage 4 EBV-associated classic Hodgkin lymphoma complicated by HLH.” – there is no such histopathological diagnosis; the diagnosis could be only classic provide the full description along with EBV tests;
We agree with your statement. Meant to say immunohistochemical stains in addition to other pathological studies. Hope this has been made more clear in edits.
what were the signs of hemophagocytosis?
HLH by meeting 6/8 HLH-2004 criterias.
Please rewrite the sentence
Stage 4 according to which classification?
Lugano staging. Information included. Thank you for the suggestions.
Why did the patient start treatment with HLH protocol and not with HL-oriented chemotherapy? In the case of lymphoma-associated HLH the treatment should be lymphoma oriented
Given high mortality associated with HLH, we felt it was appropriate to start treatment with HLH.
This is a very interesting question, as the debate still continues whether to treat the underlying cancer first or the start with HLH directed therapy (which is T cell mediated) followed by cancer directed therapy. The largest cancer series shows starting HLH like therapy followed by cancer directed therapy seems to serve better outcomes (Daver et al, Cancer 2017). In our case the patient fulfilled criteria for HLH but the work up for the cause of HLH was still unsure hence HLH directed therapy was started.
“PET-CT after two cycles…” – there is no information that PET/CT was positive before
We agree with your statement. We were unable to obtain an initial PET prior to treatment. The restaging PET showed good response to treatment.
Discussion
“Hodgkin lymphoma classically presents as lymphadenopathy around the head, neck, axilla, and mediastinum.” – it would be better to use “or” instead of “and”
Revised. Thank you for the suggestions.
“We described a case of HIV-associated Classic Hodgkin Lymphoma with primary bone marrow involvement complicated by HLH where the treatment was delayed due to difficulty in making the diagnosis.” – or because not making appropriate tests at an appropriate time?!
Our patient had primary bone marrow cHL presented with non-specific symptoms that, at the time, were likely due to infection in the setting of chronic leukopenia from HIV. Our patient improved with supportive management and empiric antibiotics. Imaging stable from prior. Initial bone marrow biopsy done suggested possible HL, but this disease is rare and ancillary test with flow was inconclusive. We believe we have completed appropriate work-up but open to suggestions.
“HIV-associated HL are also characterized by the predominance of Hodgkin/Reed-Sternberg cells whereas they are less commonly found in HL in the general population.” – what do the Authors mean?
Under morphological examination, HIV-associated HL tend to have a predominance of Reed-Sternberg compared to Hodgkin lymphoma. Hope this clears it up.
“During the diagnosis of HL, bone marrow biopsy is generally not needed due to the reliability of PET scans in diagnosing bone marrow infiltration.6 In our case, PET scan six months prior showed a lung nodule that had been biopsied twice and negative for malig-nancy each time and a stable left external iliac lymph node that was considered too small to biopsy. A bone marrow biopsy was done to assist with the diagnosis of HLH and any possible underlying bone marrow etiology. Findings were suspicious but not compelling for HL, and the flow cytometry was inconclusive.7” – please see the comments above – the whole paragraph is incorrect from the hematological point of view
Please see previous rebuttal.
“Some reports suggest that HLH-2004 may not be of benefit in patients with sepsis such as our patient” – I do not understand what the Authors mean – the patient was not diagnosed with sepsis
Given fever and worsening cytopenia, there was concern for sepsis.
“Further complicating the picture is the fact that ferritin, which is an inflammatory marker and elevated in both infection and HLH has also been shown to be markedly ele-vated in patients with untreated classic Hodgkin lymphoma without lymphadenopathy.9” – ferritin alone is not used for diagnosing HLH – I cannot see any reason for such a sentence
Included to add to discussion the ferritin is a non-specific marker that is elevated in infection, HLH, and Hodgkin lymphoma. The first being what our patient were suspected of having and the latter two being what our patient actually had.
“Interestingly, EBV viral load has been correlated with HLH severity and during our pa-tient’s second admission, when the diagnosis of HLH was apparent, his EBV viral load increased from 51K to 148K.11 This is likely because EBV plays an etiological role in the pathogenesis of HIV-associated HL.14” – the Authors mix HLH with HL – this does not make a logical sequence
EBV viral load also correlated to severity in hemophagocytic lymphohistiocytosis as per paper referenced.
